# Flaccidoxide-13-Acetate-Induced Apoptosis in Human Bladder Cancer Cells is through Activation of p38/JNK, Mitochondrial Dysfunction, and Endoplasmic Reticulum Stress Regulated Pathway

**DOI:** 10.3390/md17050287

**Published:** 2019-05-13

**Authors:** Yu-Jen Wu, Tzu-Rong Su, Guo-Fong Dai, Jui-Hsin Su, Chih-I Liu

**Affiliations:** 1Department of Nursing, Meiho University, Pingtung 91202, Taiwan; x00002180@meiho.edu.tw; 2Department of Biological Technology, Meiho University, Pingtung 91202, Taiwan; fwind101@gmail.com; 3Antai Medical Care Cooperation Antai Tian-Sheng Memorial Hospital, Pingtung 92842, Taiwan; a081002@mail.tsmh.org.tw; 4Yu Jun Biotechnology Co., Ltd., Kaohsiung 81363, Taiwan; 5National Museum of Marine Biology and Aquarium, Pingtung 94450, Taiwan; x2219@nmmba.gov.tw

**Keywords:** flaccidoxide-13-acetate, cultured soft coral *Sinularia gibberosa*, human bladder cancer, apoptosis, mitochondrial dysfunction, PERK-eIF2*α*-ATF6-CHOP pathway, p38/JNK

## Abstract

Flaccidoxide-13-acetate, an active compound isolated from cultured-type soft coral *Sinularia gibberosa*, has been shown to have inhibitory effects against invasion and cell migration of RT4 and T24 human bladder cancer cells. In our study, we used an 3-(4,5-dimethyl-2-thiazolyl)-2,5-diphenyl-2-H-tetrazolium bromide (MTT), colony formation assay, and flow cytometry to determine the mechanisms of the anti-tumor effect of flaccidoxide-13-acetate. The MTT and colony formation assays showed that the cytotoxic effect of flaccidoxide-13-acetate on T24 and RT4 cells was dose-dependent, and the number of colonies formed in the culture was reduced with increasing flaccidoxide-13-acetate concentration. Flow cytometry analysis revealed that flaccidoxide-13-acetate induced late apoptotic events in both cell lines. Additionally, we found that flaccidoxide-13-acetate treatment upregulated the expressions of cleaved caspase 3, cleaved caspase 9, Bax, and Bad, and down-regulated the expressions of Bcl-2, *p*-Bad, Bcl-x1, and Mcl-1. The results indicated that apoptotic events were mediated by mitochondrial dysfunction via the caspase-dependent pathway. Flaccidoxide-13-acetate also provoked endoplasmic reticulum (ER) stress and led to activation of the PERK-eIF2*α*-ATF6-CHOP pathway. Moreover, we examined the PI3K/AKT signal pathway, and found that the expressions of phosphorylated PI3K (*p*-PI3K) and AKT (*p*-AKT) were decreased with flaccidoxide-13-acetate concentrations. On the other hand, our results showed that the phosphorylated JNK and p38 were obviously activated. The results support the idea that flaccidoxide-13-acetate-induced apoptosis is mediated by mitochondrial dysfunction, ER stress, and activation of both the p38 and JNK pathways, and also relies on inhibition of PI3K/AKT signaling. These findings imply that flaccidoxide-13-acetate has potential in the development of chemotherapeutic agents for human bladder cancer.

## 1. Introduction

In the US, bladder cancer is the second most frequent genitourinary malignant tumor, and is a fatal disease [1]. Most human bladder cancers belong to the category of transitional cell carcinoma, which can be further divided into invasive and non-invasive bladder cancer. T2 and higher stages of muscle invasive bladder cancer have a poor prognosis and a high risk of metastasis [2,3,4]. Non-muscle-invasive bladder cancer is usually treated with transurethral resection of the bladder tumor, followed by close monitoring or Bacillus Calmette–Guérin (BCG) instillation [5]. However, the recurrence rate in patients with high-grade non-muscle-invasive bladder cancer after transurethral resection exceeds 75%, with a low survival rate [6]. Since the recurrence rate of bladder cancer is still high, and many medications currently used for bladder cancer have strong side effects [3], the development of new drugs for bladder cancer treatment remains an important issue.

Marine soft corals are rich in biologically-active substances, and have been shown to exert anti-inflammatory, anti-fungal, anti-viral, anti-cancer, and cytotoxic activities [7,8,9,10]. Cembrane -type diterpene are common secondary metabolites of marine and terrestrial organisms, and cytotoxicity is one of the major characteristics of compounds of this type [11,12,13]. Previous studies have shown that compounds extracted from soft corals, such as diterpenes, diterpenoids, and prostanoids can induce apoptosis in cancer cells, including colon cancer, oral squamous carcinoma, breast cancer, cervical cancer, hepatocellular carcinoma, bladder cancer, and melanoma cells [14,15,16,17,18,19,20]. The apoptotic process includes intrinsic and extrinsic pathways [21,22]. Studies have shown that many organelles in the cell may trigger the intrinsic pathway to induce apoptosis when stress occurs. Mitochondria and the endoplasmic reticulum (ER) are the two major organelles in which stress-induced apoptosis takes place [22,23,24]. Mitochondria provide the chemical energy required for cellular activities, and are the main components in the cell in which oxidative phosphorylation and adenosine triphosphate (ATP) synthesis take place [25]. 

Mitochondria are also involved in cell differentiation, cell signaling, and apoptosis, and have the ability to regulate cell growth and control the cell cycle [26]. During apoptosis, Bax translocates from the cytoplasm to the outer membrane of the mitochondria, and Bax and Bak oligomers form pores in the outer membrane, triggering mitochondrial dysfunction [27,28,29]. 

The functions of the ER include regulation of protein synthesis, protein folding, post-translational modification, and maintenance of intracellular calcium homeostasis [30]. When ER stress occurs, in order to relieve the stress and promote survival, cells need to initiate specific signaling pathways to limit protein synthesis, increase protein folding ability, and degrade misfolded proteins. The unfolded protein response (UPR) is a series of processes by which the ER transmits the stress signal from its lumen into the cytosol and nucleus. UPR-related genes are induced by UPR sensors, activating transcription factor 6 (ATF6) and inositol requiring enzyme 1α (IRE1-α) in response to ER stress to promote correct protein folding. Another UPR sensory protein, PKR-like ER-associated kinase (PERK), detects an overload of biosynthetic protein folding in the ER, and initiates limitation of new protein synthesis by phosphorylation of eukaryotic initiation factor 2α (eIF2α) [23,31]. Additionally, cells may prevent excessive accumulation of misfolded proteins in the ER through ER-associated degradation (ERAD). If the level of misfolded proteins does not reduce, cells will undergo apoptosis via pathways involving IRE1-α, caspase-12, and PERK/CHOP [32]. Moreover, results by scholars have shown that apoptosis in human bladder cancer cells is associated with endoplasmic reticulum stress [33,34,35].

Flacidoxide-13-acetate is a cembrane-type diterpene extracted from the cultured soft coral *Sinularia gibberosa*. Our previous study revealed that flacidoxide-13-acetate reduces cancer cell migration and invasion in T24 and RT4 human bladder cancer cell lines [36]. In this study, we investigated the mechanisms of its apoptotic and antiproliferative activities in human bladder cancer, and aimed to provide useful information to inform the development of flacidoxide-13-acetate as a new drug for bladder cancer treatment.

## 2. Results

### 2.1. Flaccidoxide-13-Acetate Inhibited Human Bladder Cancer Cell Proliferation

An 3-(4,5-dimethyl-2-thiazolyl)-2,5-diphenyl-2-H-tetrazolium bromide (MTT) and a colony formation assay were used to study the cytotoxic effects of flaccidoxide-13-acetate on T24 and RT4 bladder cancer cell lines. The MTT assay showed that the survival rates of T24 and RT4 cells were dose-dependently decreased. As shown in Figure 1A, flaccidoxide-13-acetate at a concentration of 20 µM inhibited cell survival by 40%. The colony formation assay demonstrated that the numbers of colonies also reduced with increasing flaccidoxide-13-acetate concentrations (5, 10, 15, and 20 µM) in both RT4 and T24 cells (Figure 1B). The results indicate that flaccidoxide-13-acetate has the ability to suppress cell proliferation in these two bladder cancer cell lines.

### 2.2. Flaccidoxide-13-Acetate Induced Apoptosis in RT4 and T24 Cells

The results presented in Figure 1 show that flaccidoxide-13-acetate had cytotoxic effects and inhibited cell proliferation in RT4 and T24 cell lines. We next used fluorescein isothiocyanate (FITC)-labeled annexin with propidium iodide (PI) to perform flow cytometry analysis to examine the apoptotic events in flaccidoxide-13-acetate-treated RT4 and T24 cells. As shown in Figure 2, compared with the control (3.4%), the percentages of late apoptotic cells in cell cultures treated with 5, 10, and 15 µM flaccidoxide-13-acetate were 11.0%, 11.9%, and 17.6% in T24 cells, and 17%, 20.2%, and 29.1% in RT4 cells, respectively. The dose–response results demonstrate that flaccidoxide-13-acetate induced late apoptotic events in both cell lines. 

### 2.3. Flaccidoxide-13-Acetate Initiated Mitochondrial Dysfunction in T24 and RT4 Cells

Mitochondrial dysfunction is known to be associated with apoptosis [27,28,29]. As flaccidoxide-13-acetate was found to induce a late apoptotic response in RT4 and T24 cells (Figure 2), we next examined whether flaccidoxide-13-acetate-induced apoptosis is mediated by mitochondrial dysfunction. Using Western blotting, the expressions of Bax and Bad in the cells were analyzed, and the results showed that the expression levels of Bad and Bax increased with the increase in concentrations of flaccidoxide-13-acetate, while the expression levels of p-Bad Bcl-x1, Bcl-2, and Mcl-1 decreased (Figure 3). Mitochondria are membrane-bound organelles that play an essential role in maintaining biological homeostasis, and their normal functions are controlled by Bcl-2 family proteins [37]. When cells are under stress or injured, the intracellular calcium homeostasis is disturbed, resulting in the opening of mitochondrial permeability transition pores. This results in the pro-apoptotic protein Bax being overexpressed and residing in the outer mitochondrial membrane, further forming a heterodimer with Bcl-2. It also leads to changes in the mitochondrial transmembrane potential and cytochrome *C* (Cyt *C)* release into the cytosol. Moreover, we also performed Western blotting to examine the effect of flaccidoxide-13-acetate on the activation of caspase 9, caspase 3, and PARP-1 cleavage. The results demonstrated that flaccidoxide-13-acetate treatment up-regulated expression levels of cleavage-PARP-1, cleavage-caspase 3, and cleavage-caspase 9, and down-regulated expression levels of pro-caspase 3 and pro-caspase 9. Our results showed that flaccidoxide-13-acetate induced mitochondrial dysfunction and the compounds increased active caspase level, leading to apoptosis in T24 and RT4 cells.

### 2.4. Flaccidoxide-13-Acetate Activated the p38 and JNK Pathways and Inhibited the PI3K/AKT Pathway

The mitogen-activated protein kinase (MAPK) signaling pathway plays important roles in regulating several key cell functions, including gene expression, cell proliferation, and apoptosis [38]. We used Western blotting to analyze the changes in key proteins in the MAPK signaling pathway. The results showed no significant changes in non-phosphorylated ERK, JNK, and p38, while the levels of phosphorylated p38 (*p*-p38) and JNK (*p*-JNK) were increased with the increase in flaccidoxide-13-acetate concentrations.

We next examined the PI3K/AKT signal pathway, and found that the expressions of phosphorylated PI3K (*p*-PI3K) and AKT (*p*-AKT) were decreased with the increase in flaccidoxide-13-acetate concentrations (Figure 4). We also used the p38 inhibitor (SB203580), JNK inhibitor (SP600125), and ERK inhibitor (PD98059) to validate whether apoptosis induced by flaccidoxide-13-acetate is mediated by activation of the MAPK pathway. The results demonstrated that cells pre-treated with SB203580 and SP600125, followed by incubation with flaccidoxide-13-acetate, exhibited an increase in the cell survival rate from 60% to approximately 80%, while PD98059 did not improve cell survival (Figure 4B). The abovementioned findings suggest that flaccidoxide-13-acetate-induced apoptosis in RT4 and T24 cells involves activation of the JNK and p38 pathways and inhibition of the PI3K/AKT pathway.

### 2.5. Endoplasmic Reticulum Stress is Involved in Flaccidoxide-13-Acetate-Induced Apoptosis

ER is sensitive to changes in internal and external factors, such as calcium ion concentration change and protein misfolding caused by viral infection. ER stress affects the maintenance of normal cellular functions and may even affect cell survival [39]. The ER stress response constitutes a cellular process, and may lead to unfolded protein response (UPR), ER-associated degradation (ERAD), and apoptosis [40,41]. The UPR is regulated by three ER sensors: inositol requiring enzyme 1-α (IRE1-α), protein kinase RNA-like endoplasmic reticulum kinase (PERK), and activating transcription factor 6 (ATF6). When misfolded proteins accumulate in the ER, ER chaperones (such as GRP78) release transmembrane sensor proteins PERK, IRE1-α, and ATF6, activating the UPR. In addition, PERK may regulate the signaling pathway that causes autophagy to promote cell survival [42] or increases ATF4/CHOP expression to induce apoptosis [43].

In this study, the changes in two ER transmembrane sensor proteins, PERK, ATF6, and other relevant proteins were analyzed using Western blotting. The results showed that the expressions of ATF6-f and GRP78 proteins increased in both cell lines treated with flaccidoxide-13-acetate; in addition, the levels of *p*-eIF2α and *p*-PERK, as well as ATF4 and CHOP increased with the increase in concentrations of flaccidoxide-13-acetate (Figure 5). To confirm that ER stress is involved in apoptosis, we added salubrinal, an inhibitor of eIF2α phosphorylation, into the culture, and found that cells treated with salubrinal exhibited a significantly increased survival rate (Figure 5B). The results further support that ER stress is associated with flaccidoxide-13-acetate-induced apoptosis.

## 3. Discussion

### 3.1. Flaccidoxide-13-Acetate Triggers Mitochondrial Dysfunction in T24 and RT4 Bladder Cancer Cells

The Bcl-2 family of proteins are regulators of apoptosis, and include distinct subfamilies: (1) proapoptotic proteins with a multi-BH domain (such as Bax and Bak); (2) proapoptotic proteins with a single BH3-only protein (such as Bad, Bim, and Bid), and (3) antiapoptotic members such as Bcl-2 and Bcl-xl with all four BH domains conserved (such as Bcl-2, Bcl-xl, and Mcl-1) [44]. Studies have shown that intracellular stress may trigger the intrinsic apoptotic pathway in several organelles; mitochondria and the ER are two of the major organelles involved [22,23,24]. In the presence of mitochondrial stress, Bax translocate to the outer membrane of the mitochondria and increases the membrane permeability, which causes alteration of the mitochondrial membrane potential, depolarization, and opening of mitochondrial pores [28].

We used Western blotting to analyze the changes in mitochondria-associated apoptotic proteins in flaccidoxide-13-acetate-treated T24 and RT4 cells, and found that Bax and Bad expressions were increased with increasing flaccidoxide-13-acetate concentrations, while the expressions of *p*-Bad Bcl-x1, Bcl-2, and Mcl-1 were decreased. An increased ratio of Bax/Bcl-2 provokes the release of cytochrome *C* from the mitochondrial intermembrane/intercristae spaces to the cytosol, where cytochrome binds to A-Raf-1, resulting in caspase 9 and downstream effector caspase-3 activation [45,46,47,48]. Activated caspase3 then cleaves PARP-1 to induce characteristic apoptosis changes, such as chromatin condensation and DNA fragmentation [49]. Our results imply that flaccidoxide-13-acetate treatment caused mitochondrial stress, which led to an increase in Bad expression and decreases in Bcl-2, Bcl-xl, Mcl-1, and *p*-Bad expressions, causing mitochondrial dysfunction. Additionally, the increase in Bax expression and the decrease in Bcl-2 expression resulted in an increased Bax/Bcl-2 ratio, which stimulated the release of cytochrome *C* from the mitochondrial inner membrane into the cytosol. Our results also show that flaccidoxide-13-acetate treatment up-regulated expression levels of cleavage-PARP-1, cleavage-caspase 3, and cleavage-caspase 9, and down-regulated expression levels of pro-caspase 3 and pro-caspase9. We hypothesize that the release of cytochrome *C* causes procaspase 9 to self-hydrolyze into activated caspase 9, which in turn activates caspase 3 and cleaves PARP into its cleaved form. The process subsequently triggered apoptotic events, such as DNA fragmentation and chromosome condensation [49]. These results imply that flaccidoxide-13-acetate-induced apoptosis in both T24 and RT4 human bladder cancer cells are related to the pathway associated with activated mitochondrial apoptosis.

### 3.2. Flaccidoxide-13-Acetate Induces Apoptosis through Activation of the p38 and JNK Signaling Pathways and Inhibition of the PI3K/AKT Pathway

The MAPK pathway regulates many stress-induced physiological processes, including cell differentiation, cell growth, and apoptosis [50,51]. Three well-characterized subfamilies of the MAPK pathway are mediated by JNKs, p38 kinases, and ERKs, respectively. Among these MAPKs, activation of JNKs and p38 kinases occurs in response to various intrinsic and extrinsic stimuli [51,52]. Study has shown that intracellular stress may induce p38 pathway activation and promote apoptosis by activation of p53 and inhibition of Bcl-2 expression [53]. ERK is important for cell proliferation and survival, and can be activated by mitotic stimuli, such as growth factors and cytokines [54]. Phosphorylated ERK has been shown to promote cell proliferation and inhibit pro-apoptotic signaling [55], and constitutive activation and overexpression of ERK are often observed in many cancer cells [56]. In this study, Western blotting showed that the total expressions of ERK, JNK, and AKT were found to be unchanged in T24 and RT4 cells treated with various concentrations of flaccidoxide-13-acetate, while the expressions of phosphorylated p38 and JNK increased significantly (Figure 4). When specific inhibitors of p38, JNK, and ERK were used, the results demonstrated that both the p38 inhibitor (SB203580) and the JNK inhibitor SP600125, but not the ERK inhibitor (PD98059), increased the cell viability, suggesting that flaccidoxide-13-acetate-induced apoptosis in RT4 and T24 cells is at least partly mediated by the p38 and JNK pathways.

The PI3K/AKT signaling pathway has been found to be activated in several types of cancer, and is known to regulate cell survival, proliferation, differentiation, and apoptosis [57]. Several downstream targets of AKT, such as GSK-3b, Bad, and NF-κB, are known to directly or indirectly regulate apoptosis in many cancers [58,59]. Phosphorylation of Bad alters its binding ability to Bcl-2 and Bcl-xl by increasing its association with 14-3-3, resulting in loss of the ability of Bad to heterodimerize with survival proteins Bcl-2 and Bcl-xl [60,61]. Therefore, downregulation of the PI3K/AKT signaling pathway has been found to induce apoptosis in several studies [62,63]. We showed that RT4 and T24 cells under flaccidoxide-13-acetate treatment exhibited decreases in the expressions of phosphorylated PI3K and AKT expression, which indicates that flaccidoxide-13-acetate-induced apoptosis occurs through inactivation of the PI3K/AKT signaling pathway.

### 3.3. Flaccidoxide-13-Acetate-Induced Apoptosis Occurs Partially via Initiation of ER Stress in T24 and RT4 Cells

When cells are under ER stress, GRP78, which is involved in protein folding, initiates the release of transmembrane proteins PERK and IRE1-α, and activates transcription factor ATF6. PERK signaling has been demonstrated to utilize autophagy as a survival strategy [42], or cause apoptosis through upregulation of ATF4 and proapoptotic transcription factor CCAA/enhancer binding protein (C/EBP) homologous protein (CHOP) [43]. Upon ER stress, transcription and protein synthesis are reduced due to eIF2α being phosphorylated by *p*-PERK, and the ATF6 and IRE1-α signaling pathways induce the expression of ER chaperones [64,65,66]. When ER stress is prolonged, PERK is activated and phosphorylates eIF2α [67,68]; phosphorylated eIF2α subsequently activates ATF4, which targets the expression of apoptotic effector, CHOP [69]. Apoptosis is related to the expression of CHOP; expression of ATF4 is also positively correlated with CHOP. Thus, when ATF4 induces downstream CHOP protein expression, the cells will progressively switch from autophagy to apoptosis [70].

Our results demonstrate that the expressions of GRP78 proteins and ATF6 increased in both cell lines treated with flaccidoxide-13-acetate. In addition, the levels of *p*-PERK, *p*-eIF2α, and ATF4 as well as CHOP were increased with increase in concentration of flaccidoxide-13-acetate treatment (Figure 5A). To confirm that ER stress is involved in apoptosis, we added salubrinal, an inhibitor of eIF2α phosphorylation, into the culture, and found that cells treated with salubrinal exhibited a significantly increased survival rate (Figure 5B). 

These results imply that the apoptosis induced by flaccidoxide-13-acetate is partially mediated by the PERK-eIF2α-ATF4-CHOP pathway. These results are similar to the outcomes of Toeh et al.’s study [71]. When the function of the endoplasmic reticulum is disturbed, it will lead to ER stress that activates the expression of PERK and ATF6, then induces CHOP up-regulation of the PERK/ATF4/CHOP pathway and causes cell apoptosis

## 4. Material and Methods

### 4.1. Reagents

Flaccidoxide-13-acetate was isolated from cultured-type soft coral *Sinularia gibberosa* by Dr. Jui-Hsin Su. Dulbecco’s modified Eagle’s medium (DMEM), fetal bovine serum (FBS), and phosphate-buffered saline (PBS), were purchased from Biowest (Nuaillé, France). Polyvinylidene difluoride (PVDF) membranes were purchased from Millipore (Bellerica, MA, USA). Protease inhibitor cocktail, dimethyl sulfoxide (DMSO), salubrinal, and goat anti-rabbit and horseradish peroxidase-conjugated immunoglobulin (Ig) G were obtained from Sigma (St. Louis, MO, USA). Cell extraction buffer was acquired from BioSource International (Camarillo, CA, USA). An annexin V-FITC/PI apoptosis detection kit was purchased from Pharmingen (San Diego, CA, USA). The enhanced chemiluminescence (ECL) Western blotting reagents were obtained from Pierce Biotechnology (Rockford, IL, USA). Cytochrome *C* releasing apoptosis assay kit was purchased form Biovision (Milpitas, CA, USA).

### 4.2. Cell Culture and Drug Treatment

Human bladder cancer RT4 and T24 cell lines were obtained from the Taiwan Food Industry Research and Development Institute (Hsinchu, Taiwan). The cell lines were cultured in Dulbecco’s modified Eagle’s medium (DMEM) supplemented with 10% fetal bovine serum (FBS), 100µg/mL streptomycin, and 100 units/mL penicillin in a humidified 5% CO_2_ incubator at 37 °C. Cells were treated with various concentrations of flaccidoxide-13-acetate and harvested after 24 h of incubation. DMSO was added to the control group, and cultured for 24 h for subsequent studies. All experiments were performed three times to determine their reproducibility.

### 4.3. Cell Viability Assay

The effects of flaccidoxide-13-acetate on the T24 and RT4 cell lines were evaluated by MTT assay. Cells (1 × 10^5^ cells/well) were seeded in 96 well plates. The cells were treated with various concentrations of flaccidoxide-13-acetate (5, 10, 15, 20, and 25µM). After 24 h of incubation, MTT solution (0.5 mg/mL in PBS) was added to each well. The plates were incubated for 4 h at 37 °C, after which the culture medium was removed and the cells were dissolved in 200 μL DMSO. The absorbance was measured at 595 nm using a microplate ELISA reader (Bio-Rad, Hercules, CA, USA) and DMSO was used as the control. Samples were analyzed and all experiments were repeated three times.

### 4.4. Flow Cytometric Assay

T24 and RT4 cells were seeded onto 5 cm petri dishes, treated with different concentrations of flaccidoxide-13-acetate for 24 h. The cells were then collected and fixed in 70% cold ethanol at 4 °C overnight. The cells were subsequently stained with 10 μg/mL Annexin V–FITC and 5 μg/mL propidium iodide (PI) for 30 minutes at 37 °C. Apoptosis processes induced by flaccidoxide-13-acetate were analyzed using a FACScalibur flow cytometer and Cell-Quest software (Becton-Dickinson, Mansfield, MA, USA).

### 4.5. Colony Formation Assay

T24 and RT4 cells were seeded in 24 well plates (2000 cells/well) and incubated for 24 h. The cells were treated with various concentrations (5, 10, 15, and 20 μM) of flaccidoxide-13-acetate in 2 mL of serum complete media and incubated for 10 days. The colonies were then washed with PBS and fixed with methanol for 15 min and stained with 0.15% crystal violet. The colonies were counted and scanned using a high-resolution scanner Scan Maker 9800XL (MiCROTEK, Hsinchu, Taiwan).

### 4.6. Antibody and Western Blot Assay

Rabbit anti-human ERK, *p*-ERK, JNK, *p*-JNK, GRP78, ATF4, and cleaved-ATF6 antibodies were purchased from ProteinTech Group (Chicago, IL, USA). Rabbit anti-human antibodies against AKT, *p*-AKT, PI3K, *p*-PI3K, Mcl-1, Bad, *p*-Bad, Bcl-xl, Bcl-2, Bax, p38, *p*-p38, PERK, *p*-PERK, elF2α, *p*-elF2α, pro-caspase 3, cleaved caspase 3, pro-caspase 9, cleaved caspase 9, cytochrome *C*, and CHOP were obtained from Cell Signaling Technology (Danvers, MA, USA). Rabbit anti-human β-actin antibodies were obtained from Sigma (St Louis, MO, USA). Cytosolic cytochrome *C* were separated using a cytochrome *C* releasing apoptosis assay kit (Biovision, Milpitas, CA, USA). 

The flaccidoxide-13-acetate treated sample and DMSO treated control samples (total protein 25 μg) were separated by 12.5% SDS-PAGE, and the proteins on the gel were transferred to a PVDF membrane. The membrane containing transferred protein was blocked in PBS buffer and incubated with primary antibody at 4 °C overnight, followed by secondary antibodies (goat anti-rabbit or goat anti-mouse and horseradish peroxidase conjugate, 1:5000 dilution in 2% dehydrated skim milk) for 2 h at 4 °C. The signals were detected with an enhanced chemiluminescence detection kit.

### 4.7. Inhibitor Assessment

In order to further determine the effects of p38, ERK, and JNK on flaccidoxide-13-acetate-induced cell proliferation arrest, a total of 1 × 10^5^ cells were seeded in a 24 well plate and pre-incubated for 2 h with specific inhibitors for p38 (SB2203580), JNK (SP600125), and ERK (PD98059) prior to flaccidoxide-13-acetate administration. Afterwards, the cell viability rate was determined by MTT assay.

### 4.8. Statistical Analysis 

The results of the MTT assay and colony formation assay were subjected to Student’s test (Sigma-Stat2.0, San Rafael, CA, USA). Results with *p* < 0.05 were considered statistically significant.

## 5. Conclusions

In this study, we demonstrated that flaccidoxide-13-acetate induces apoptosis in RT4 and T24 bladder cancer cells, and the process is mediated by induction of mitochondrial dysfunction and activation of ER stress, which also involves initiation of the p38 and JNK pathways and inhibition of the PI3K/AKT pathway (Figure 6). Our findings revealed that flaccidoxide-13-acetate-induced apoptosis in bladder cancer cells occurs via multiple pathways. Further study of the underlying mechanism is underway in our laboratory to identify specific targets of flaccidoxide-13-acetate in bladder cancer cells.

## Figures and Tables

**Figure 1 marinedrugs-17-00287-f001:**
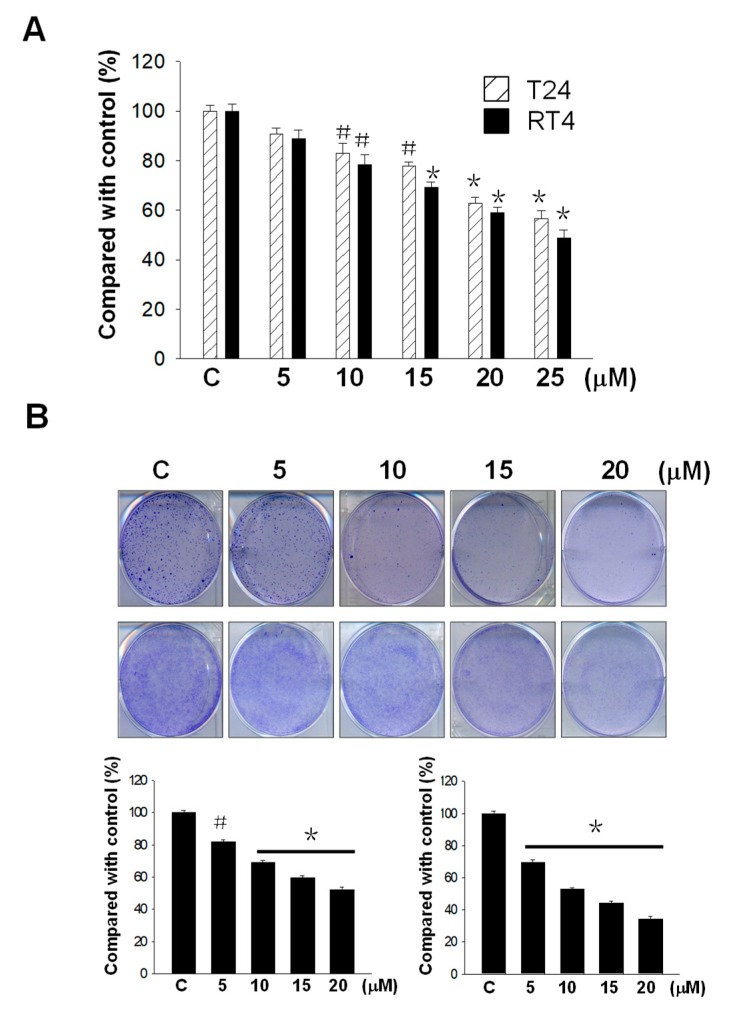
(**A**) Cytotoxic effects of flaccidoxide-13-acetate (0, 5, 10, 15, 20, 25 µM) on T24 and RT4 cell lines. T24 and RT4 cells were incubated with the indicated flaccidoxide-13-acetate concentrations for 24 h, and the cell numbers were assessed using an 3-(4,5-dimethyl-2-thiazolyl)-2,5-diphenyl-2-H-tetrazolium bromide (MTT) assay. The results were obtained from three individual experiments. (**B**) Effects of flaccidoxide-13-acetate on colony formation in RT4 and T24 cell lines. The cells were treated with various (5, 10, 15, and 20 µM) concentrations of flaccidoxide-13-acetate and cultured for 10 days. The numbers of colonies were counted and the results were normalized to a culture without flaccidoxide-13-acetate treatment (100%). Data are presented as mean± S.D. of triple replicate experiments. (^#^
*p* < 0.05; * *p* < 0.01, compared with the control.)

**Figure 2 marinedrugs-17-00287-f002:**
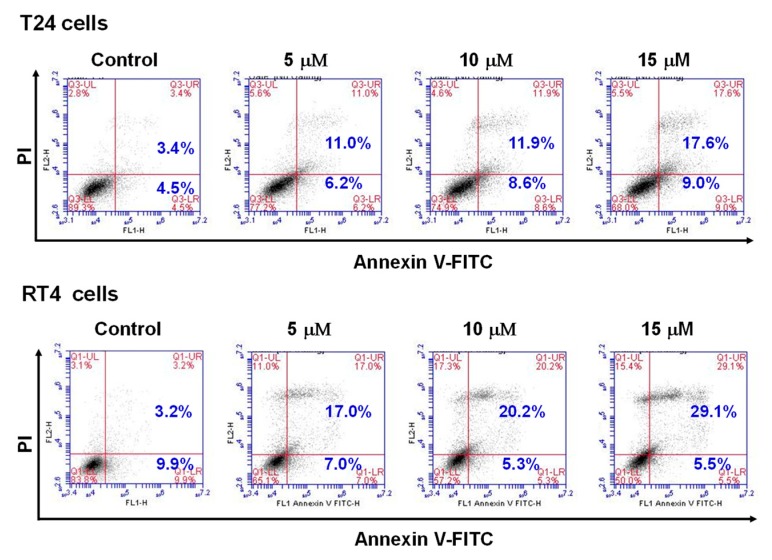
Detection of apoptosis in flaccidoxide-13-acetate-treated RT4 and T24 cells by flow cytometry analysis.

**Figure 3 marinedrugs-17-00287-f003:**
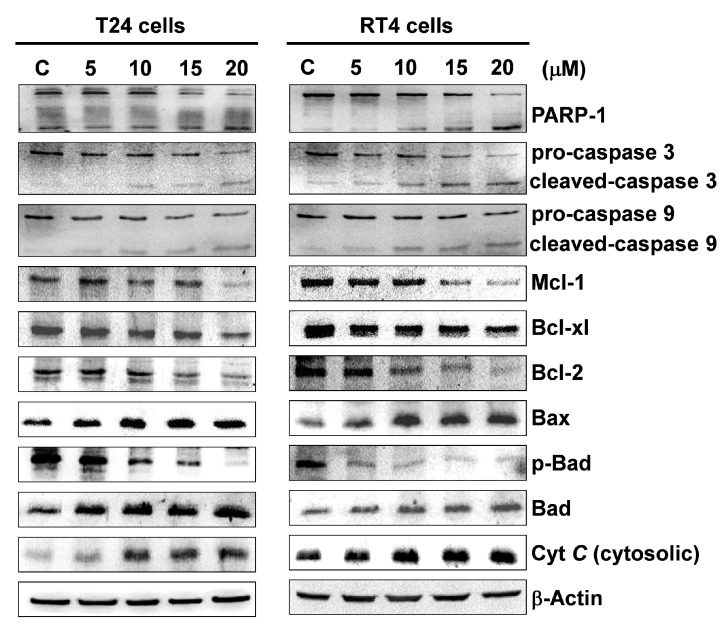
Western blotting analysis of the expressions of Bcl-2 family proteins and cytochrome *C* after flaccidoxide-13-acetate-treated in RT4 and T24 cells. With increased flaccidoxide-13-acetate concentrations, the expressions of Bax, Bad, and Cyt *C* were increased, but Mcl-1, Bcl-xl, Bcl-2, and *p*-Bad were decreased.

**Figure 4 marinedrugs-17-00287-f004:**
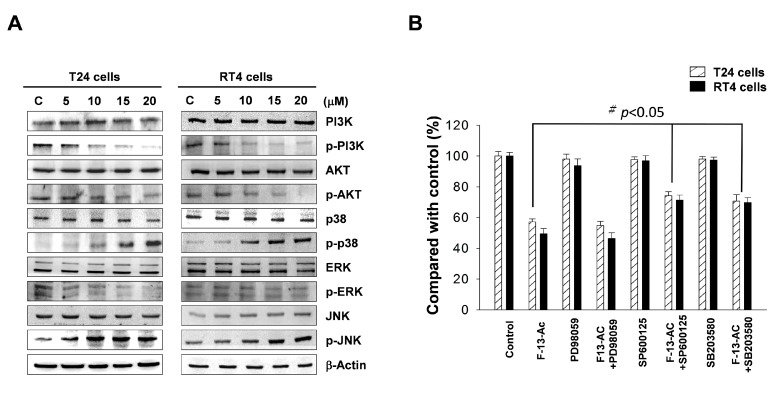
(**A**) Western blotting analysis of the expressions of PI3K/AKT and mitogen-activated protein kinase (MAPK) pathway-related proteins in RT4 and T24 cells after administration of flaccidoxide-13-acetate treatment. (**B**) Flaccidoxide-13-acetate-induced apoptosis in T24 and RT4 cells is mediated by activation of p38 and JNK pathway using ERK-, JNK-, and p38-specific inhibitors. (F-13-AC: flaccidoxide-13-acetate). Data are presented as mean± S.D. of triple replicate experiments. (^#^
*p*< 0.05; * *p* < 0.01, compared with the control).

**Figure 5 marinedrugs-17-00287-f005:**
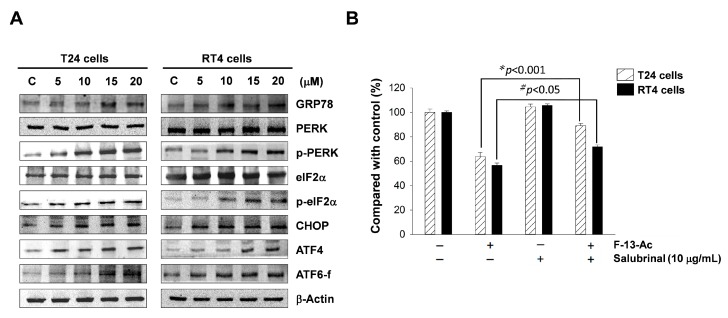
(**A**) Western blotting analysis of ER stress-related proteins in T24 and RT4 cells treated with flaccidoxide-13-acetate. (**B**) Effect of eIF2α phosphorylation inhibitor salubrinal on the cell survival of the two cell lines treated with flaccidoxide-13-acetate.

**Figure 6 marinedrugs-17-00287-f006:**
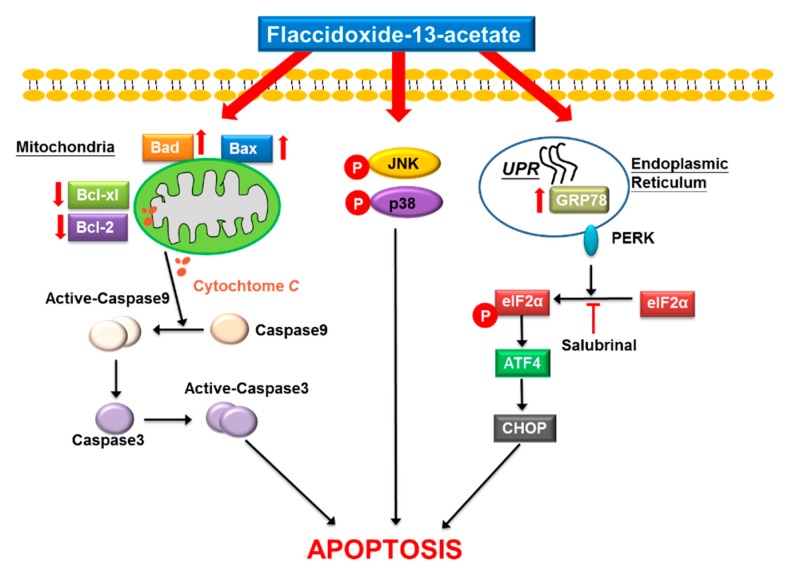
Flaccidoxide-13-acetate-induced apoptotic pathway in bladder cancer cells. The anti-cancer effect of flaccidoxide-13-acetate is mediated by the induction of mitochondria dysfunction and ER stress signaling pathways, also involving initiation of the p38 and JNK pathways.

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
