# Peer review of "Flaccidoxide-13-Acetate-Induced Apoptosis in Human Bladder Cancer Cells is through Activation of p38/JNK, Mitochondrial Dysfunction, and Endoplasmic Reticulum Stress Regulated Pathway"

_marinedrugs, 2019, doi:10.3390/md17050287_

Round 1
Reviewer 1 Report
In this papers authors have studies the action of Flaccidoxide‐13‐acetate, an active compound isolated from cultured‐type soft coral sinularia gibberosa, against RT4 and T24 human bladder cancer cells.
Based on the results presented authors conclude that flaccidoxide‐13‐acetate‐induced apoptosis mediated by the mitochondrial dysfunction, ER stress, activation of both the p38/JNK pathways, and also relies on inhibition of PI3K/AKT signaling, suggesting a potential in the development of chemotherapeutic agents for human bladder cancer.
Actually, authors demonstrate that the effects of cytoxicity and against invasion and cell migration are due to apoptotic events in both cell lines, as convincingly evidenced by the upregulated expressions of cleaved caspase 3, cleaved caspase 9, Bax, Bad, and down‐regulated the expressions of Bcl‐2, p‐Bad, Bcl‐x1, and Mcl‐1.
Similarly, the rationale is supported also by the demonstration of endoplasmic reticulum (ER) stress and activation of the PERK‐eIF2a‐ATF6‐CHOP pathway and involvement of the PI3K/AKT signal pathway.
I have only few concerns.
Since the effects were dose-dependent and the most effective concentration was 25 µM, that is also approximately the IC50 value of cytotoxic experiments, I wonder why all the other experiments were conducted up to 15 and 20 µM at maximum.
In addition, the reference list does not seem very up to date, since the most recent reference is only one and from 2017, and then three from 2015 and so on.
Author Response
Reviewer 1
In this papers authors have studies the action of Flaccidoxide‐13‐acetate, an active compound isolated from cultured‐type soft coral sinularia gibberosa, against RT4 and T24 human bladder cancer cells.
Based on the results presented authors conclude that flaccidoxide‐13‐acetate‐induced apoptosis mediated by the mitochondrial dysfunction, ER stress, activation of both the p38/JNK pathways, and also relies on inhibition of PI3K/AKT signaling, suggesting a potential in the development of chemotherapeutic agents for human bladder cancer.
Actually, authors demonstrate that the effects of cytoxicity and against invasion and cell migration are due to apoptotic events in both cell lines, as convincingly evidenced by the upregulated expressions of cleaved caspase 3, cleaved caspase 9, Bax, Bad, and down‐regulated the expressions of Bcl‐2, p‐Bad, Bcl‐x1, and Mcl‐1.
Similarly, the rationale is supported also by the demonstration of endoplasmic reticulum (ER) stress and activation of the PERK‐eIF2a‐ATF6‐CHOP pathway and involvement of the PI3K/AKT signal pathway.
I have only few concerns.
Since the effects were dose-dependent and the most effective concentration was 25 µM, that is also approximately the IC50 value of cytotoxic experiments, I wonder why all the other experiments were conducted up to 15 and 20 µM at maximum.
Respond: Thank for reviewer’s suggestion. Because cells have approximately 50% mortality at a concentration of 25 mM, subsequent experiments were performed to avoid the result of cell necrosis, and the highest concentration was used up to 20 mM in subsequent experiments.
In addition, the reference list does not seem very up to date, since the most recent reference is only one and from 2017, and then three from 2015 and so on.
Respond: Thank for reviewer’s suggestion. We have added reference 8,9,10,33,34 and 35 in the manuscript.
Reviewer 2 Report
Most of my comments the authors and the Editor will find in the attached copy of the manuscript. Generally, the publication is well written.
My major concern is about the discussion part - in my opinion, this part just repeats the results and doe not discuss them in e.g. comparison to other flavonoids. Some extensive (also in the introduction) descriptions of molecular pathways are unnecessary but would be acceptable if they were accompanied by a proper discussion.
Some additional, minor issues that I would like to note - error bars description (SD, SEM etc.), number of assays repeats (e.g. each blot was performed only once ?).

Author Response
Reviewer 2
Most of my comments the authors and the Editor will find in the attached copy of the manuscript.
Respond: Revisions were done according to reviewer's comments.
Generally, the publication is well written.
My major concern is about the discussion part - in my opinion, this part just repeats the results and doe not discuss them in e.g. comparison to other flavonoids. Some extensive (also in the introduction) descriptions of molecular pathways are unnecessary but would be acceptable if they were accompanied by a proper discussion.
Respond: Thank for reviewer’s suggestion. We have modified some of the discussion part.
Some additional, minor issues that I would like to note - error bars description (SD, SEM etc.), number of assays repeats (e.g. each blot was performed only once ?).
Respond: The immunostaining analysis will perform three repetitions. In the manuscript, we will select an immunostaining analysis result as a representative.
Round 2
Reviewer 2 Report
Generally, my suggestions were introduced and the discussion is much more useful know.
My only concern is about WB and FACS analysis (Annexin V)
"Respond: The immunostaining analysis will perform three repetitions. In the manuscript, we will select an immunostaining analysis result as a representative."
If three replicates were done, the authors should analyze all of them, show the results of the analyses as a graph accompanying the representative blot, dot-plot. Something like Fig. 1 B
Author Response
Respond:
Thank for reviewer's suggestion. We performed western blotting analysis on RT4 and T24 cell lines, and the results of the experiment were 66 item. If the Western blot is to be presented as shown in Figure 1B, the graph will be too complicated. Therefore, we present the results in the text as a differential expression of proteins.